# Writing and reading antiferromagnetic Mn$_2$Au by Néel spin-orbit torques and large anisotropic magnetoresistance

S.Yu. Bodnar[1], L. Šmejkal[1,2,3], I. Turek[3], T. Jungwirth[2,4], O. Gomonay[1], J. Sinova[1], A.A. Sapozhnik[1], H.-J. Elmers[1], M. Kläui [1] & M. Jourdan[1]

Using antiferromagnets as active elements in spintronics requires the ability to manipulate and read-out the Néel vector orientation. Here we demonstrate for Mn$_2$Au, a good conductor with a high ordering temperature suitable for applications, reproducible switching using current pulse generated bulk spin-orbit torques and read-out by magnetoresistance measurements. Reversible and consistent changes of the longitudinal resistance and planar Hall voltage of star-patterned epitaxial Mn$_2$Au(001) thin films were generated by pulse current densities of $\simeq 10^7$ A/cm$^2$. The symmetry of the torques agrees with theoretical predictions and a large read-out magnetoresistance effect of more than $\simeq 6\%$ is reproduced by ab initio transport calculations.

[1] Institut für Physik, Johannes Gutenberg-Universität, Staudinger Weg 7, 55128 Mainz, Germany. [2] Institute of Physics, Academy of Sciences of the Czech Republic, Cukrovarnicka 10, 162 00 Praha 6, Czech Republic. [3] Department of Condensed Matter Physics, Faculty of Mathematics and Physics, Charles University, Ke Karlovu 5, 12116 Praha 2, Czech Republic. [4] School of Physics and Astronomy, University of Nottingham, University Park, Nottingham NG7 2RD, UK. Correspondence and requests for materials should be addressed to M.J. (email: Jourdan@uni-mainz.de)

Antiferromagnets (AFMs) are magnetically ordered materials which exhibit no net moment and thus are insensitive to magnetic fields. Antiferromagnetic spintronics[1] aims to take advantage of this insensitivity for enhanced stability, while at the same time active manipulation up to the natural THz dynamic speeds of AFMs[2] is possible, thus combining exceptional storage density and ultrafast switching. However, the active manipulation and read-out of the Néel vector (staggered moment) orientation is challenging. Recent predictions have opened up a path based on a new spin-orbit torque[3], which couples directly to the Néel order parameter. This Néel spin-orbit torque was first experimentally demonstrated in a pioneering work using semimetallic CuMnAs[4].

For the key application operations of reading and writing in AFMs, different approaches have been previously put forward. Initial experiments on spin-valve structures with an AFM as the active layer manipulated the Néel vector by an exchange-spring effect with a ferromagnet (FM) and read-out via tunneling-anisotropic magnetoresistance (T-AMR) measurements[5]. Other related experiments were based on the same effect[6], or on a FM to AFM phase transition[7]. However, the most promising approach is to use current-induced spin-orbit torques for switching the Néel vector. It exhibits superior scaling and its counterpart in FMs is already established and considered among the most efficient switching mechanisms for memory applications[8,9].

Only two compounds, CuMnAs and $Mn_2Au$, are known to provide at room temperature the collinear commensurate antiferromagnetic order and specific crystal structure, which is predicted to result in the staggered spin accumulation in the sublattice structure, leading to bulk Néel spin-orbit torques allowing for current-induced switching of the Néel vector[3].

Semimetallic CuMnAs was grown previously by molecular beam epitaxy (MBE) with a Néel temperature of $\simeq 500\,K$[10] and current-induced switching of these samples was recently demonstrated for the first time[4,11]. However, for spintronics applications the compound $Mn_2Au$ provides several advantages,

as it is a good metallic conductor and does not contain toxic components. Furthermore, its magnetic ordering temperature is well above $1000\,K$[12], providing the necessary thermal stability for applications. $Mn_2Au$ shows a simple antiferromagnetic structure with the collinear magnetic moments in the (001) plane[12–14]. Thin film samples were previously grown in (101) orientation by MBE[15] and $Fe/Mn_2Au(101)$ bilayers showed AMR effects of up to 2.5% in a 14 T rotating magnetic field[16].

While $Mn_2Au$ was the first compound for which current-induced internal staggered spin-orbit torques were predicted[3], corresponding experimental evidence has been missing. Here we report current-induced Néel vector switching in $Mn_2Au(001)$ epitaxial thin films, which is easily read-out by a large AMR. We compare the experimental results with calculations of the AMR from ab initio theory, from which we conclude an extrinsic disorder related origin of the exceptionally large magnetoresistance.

## Results

**$Mn_2Au$ samples preparation.** Our $Al_2O_3(Substrate)/Ta(10\,nm)/Mn_2Au(75\,nm)/Ta(3\,nm)$ samples were prepared by radio frequency sputtering from a single stoichiometric target and structurally and magnetically characterized as described elsewhere[17]. By x-ray diffraction, we demonstrated that the $Mn_2Au$ thin films grow with the (001) axis perpendicular to the thin film surface. The in-plane orientation is given by the epitaxial relation with the Ta(001) buffer layer, which results in the [100] direction of the $Mn_2Au$ thin films aligned parallel to the [100] direction of the epitaxial Ta layer. These samples were patterned into a star structure as shown in Fig. 1a. This geometry allows for electric writing of the Néel vector orientation by pulsing currents along the two perpendicular directions $I_{pulse}^1$ and $I_{pulse}^2$ and for electric read-out by measuring either the transversal resistivity $\rho_{xy}$, i.e., the Planar Hall Effect (PHE), or the longitudinal resistivity $\rho_{long}$, corresponding to the AMR of the samples. Depending on the in-plane orientation of the patterned structure, the pulse currents can be sent along different crystallographic directions, i.e., along

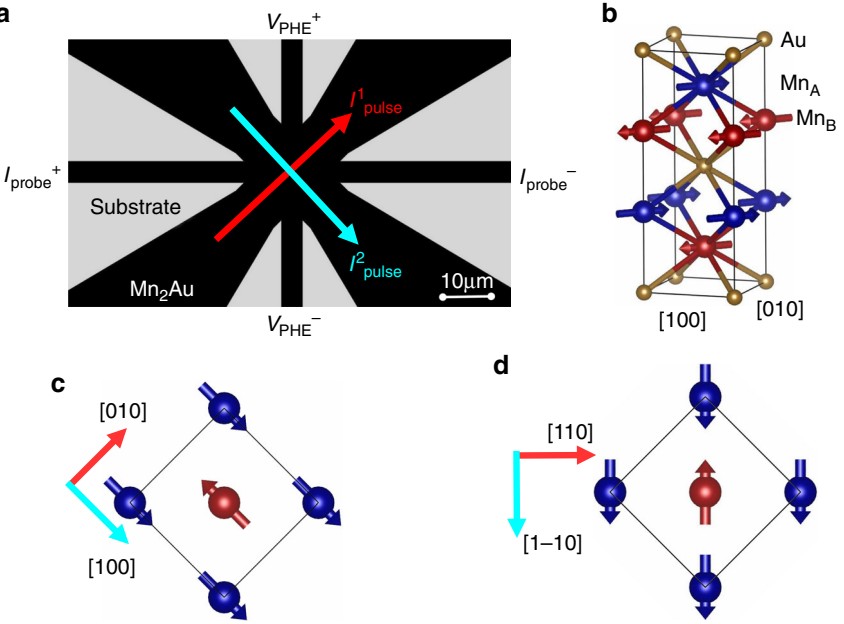

**Fig. 1** Sample layout. **a** Star pattern used for the current-induced Néel vector manipulation experiments with current pulse directions $I_{pulse}^{1/2}$ and probing contacts for Planar Hall Effect (PHE) measurements indicated. **b** shows the crystal structure of $Mn_2Au$ with arbitrarily selected in-plane orientation of the magnetic moments. **c** shows the in-plane orientation of the epitaxial $Mn_2Au$ thin films, which corresponds to the star pattern shown in **a**. $I_{pulse} \parallel [010]$ is expected to rotate the magnetic moments in the indicated directions. **d** 45° in-plane rotation of the star pattern. $I_{pulse} \parallel [110]$ is expected to rotate the magnetic moments in the indicated directions

[100] or along [110] (45° in-plane rotation of the star pattern, Fig. 1b–d).

**Relation of AMR and PHE.** The AMR of a single domain sample is given by

$$\text{AMR}_{hkl} = \frac{\rho_{\text{long}}(\phi = 0^\text{o}) - \rho_{\text{long}}(\phi = 90^\text{o})}{\left(\rho_{\text{long}}(\phi = 0^\text{o}) + \rho_{\text{long}}(\phi = 90^\text{o})\right)/2} = \frac{\Delta\rho_{\text{long}}}{\overline{\rho}_{\text{long}}}, \quad (1)$$

where $\rho_{\text{long}}$ is longitudinal resistivity, $\phi$ is the angle between the Néel vector and current direction and [hkl] is the Néel vector orientation in the basis of the tetragonal conventional unit cell (Fig. 1b). The PHE usually observed in ferromagnetic materials scales with the AMR and shows a dependence on the angle $\phi$ given by[18,19]:

$$\rho_{xy} = \Delta\rho_{\text{long}} \sin\phi \cos\phi \quad (2)$$

Thus also in AFMs, $\rho_{xy}$ has its maximum value and changes sign if $\phi$ switches from +45° to −45°.

**Switching and read-out.** For technical reasons the applicable pulse length for switching was limited to a minimum of 1 ms. This allows us using an oscilloscope to monitor the time-dependent sample resistivity during the application of the current pulses (Fig. 2). By comparison with the temperature dependence of the resistivity of our Mn$_2$Au thin films obtained from separate measurements, this allows to study the current pulse induced heating effects. As shown in Fig. 2, significant heating with local temperatures up to 300 °C is obtained for the highest possible current densities not resulting in sample destruction.

Current-induced Néel vector switching was only observed applying pulse current densities associated with notable heating. As single pulses resulted in very small changes of the read-out signals only, trains of 100 current pulses with a pulse length of 1 ms and a delay between the pulses of 10 ms were applied. This relatively long delay between the pulses was chosen to avoid the destruction of the sample by excessive accumulated heating. After a pulse train thermal relaxation on a time scale of 1 s was observed. Thus the read-out was performed with a delay of 10 s.

The magnetic state of a sample probed this way is long-term stable, i.e., no changes of the read-out signals were observed within the probed time scales of up to 1 h.

Figure 3 shows the transversal resistivity $\rho_{xy}$ versus the number of applied pulse trains. First, a pulse current density of $1.4 \times 10^7$ A/cm$^2$ was applied along the [1$\overline{1}$0] direction, resulting in a small change of the corresponding Hall voltage. Without reaching saturation after 50 pulse trains the pulse current direction was switched to [110], resulting in a reversal of the corresponding change of the transversal resistivity. This sequence could be reproduced several times. Increased pulse current densities of $1.7 \times 10^7$ and $1.8 \times 10^7$ A/cm$^2$ resulted in larger changes of the corresponding Hall voltages. Those current densities are about 1 order of magnitude smaller than predicted by Železný et al.[3,20], which could be related to thermal activation processes enabled by the above mentioned current pulse induced heating. By increasing the number of pulse trains applied along the [110] direction to 500, a trend towards saturation of the Hall voltage was obtained.

Internal field like spin-orbit torques are expected to generate reversible switching between distinct stable states if the current is injected along biaxial easy directions[3,21]. However, we observed reversible switching to stable states for pulse currents along both the crystallographic [110] and [100] axes (rotated star pattern). Thus we conclude that the in-plane magnetic anisotropy of our Mn$_2$Au thin films is weak. This is consistent with our calculations of the magnetocrystalline anisotropy energy (MAE), which is almost negligible within the (001) plane (Methods section).

An example of the resulting changes of the transversal and longitudinal resistivities generated by pulse currents along the [100] and [010] directions is displayed in Fig. 4. The inset of Fig. 4a shows that it is possible to apply current pulses until a large degree of saturation of the induced longitudinal resistivity changes is obtained. However, this is achieved for current densities that result for prolonged injection of pulses in the destruction of the samples. Thus to allow for an increased number of current pulse cycles with different polarities this regime has to be avoided. The current density at which eventually the sample is destroyed varies by a factor of ≃2 from sample to sample. The main upper panel (Fig. 4a) shows the longitudinal

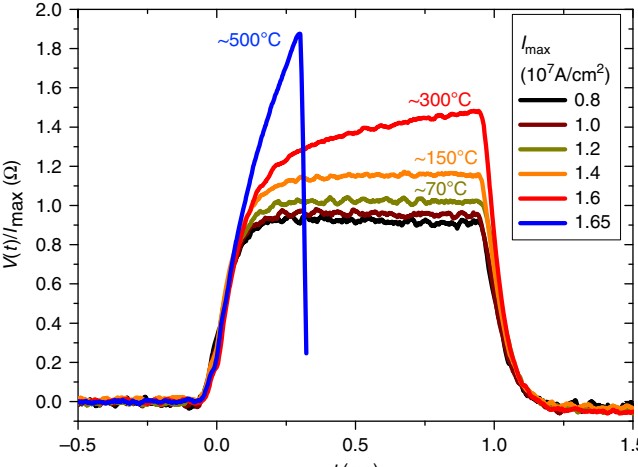

**Fig. 2** Current pulse induced heating. Time dependent normalized voltage $V(t)/I_{\text{max}}$ measured between the contacts labled $I_{\text{probe}}^+$ and $V_{\text{PHE}}^+$ during application of a current pulse $I_{\text{pulse}}^1$ with different maximum values $I_{\text{max}}$ (Fig. 1). The current pulse induced temperature of our sample is estimated from a comparison with the Mn$_2$Au temperature dependent resistivity R(T)[17] extrapolated above room temperature

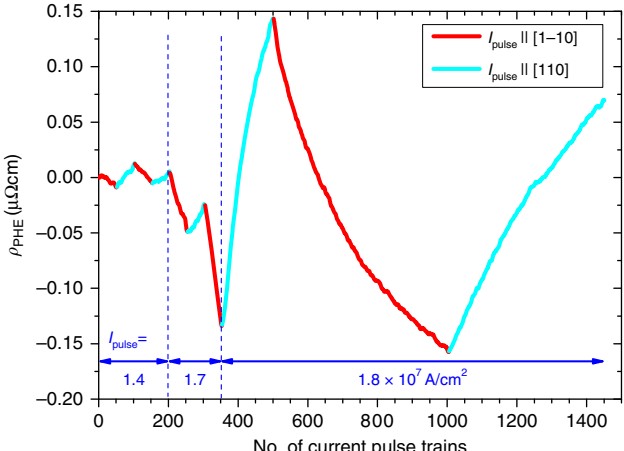

**Fig. 3** Transversal resistivity. Probed transversal resistivity (DC probing current density 10$^4$ A/cm$^2$) vs. number of applied pulse trains along different directions. The crystallographic direction in which the current pulses were injected is indicated by the cyan and red color of the data points. The pulse current density was increased from $1.4 \times 10^7$ to $1.8 \times 10^7$ A/cm$^2$ as indicated in the graph

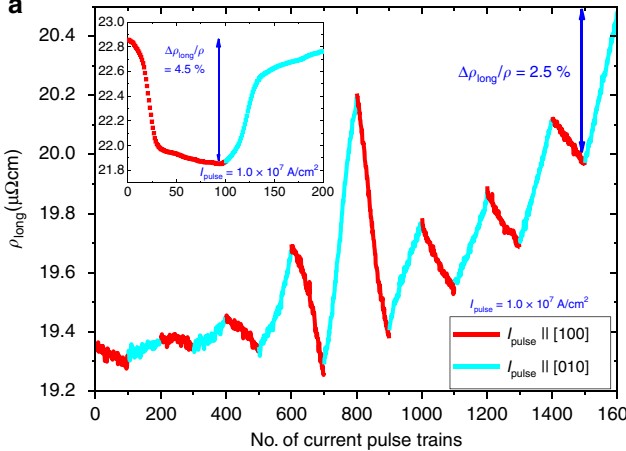

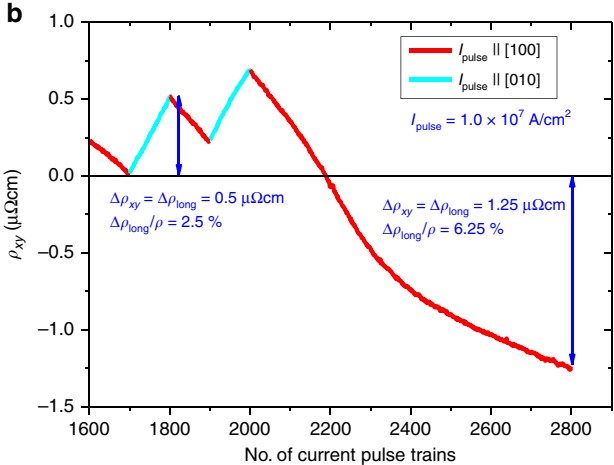

**Fig. 4** Longitudinal and transversal resistivities. **a** Longitudinal resistivity (DC probing current density $10^4$ A/cm$^2$) vs. number of applied pulse trains. The inset shows the longitinal resisitvity of another sample. In this case saturation of the magnetoresistance could be obtained for a few repetitions before the sample broke. **b** Transversal resistivity of the same sample as shown in the main panel of **a** vs. number of applied pulse trains. The crystallographic direction in which the current pulses were injected is indicated by the cyan and red color of the data points

resistivity probed after each of the first 1600 pulse trains consisting of 100 pulses each with a current density $1.0 \times 10^7$ A/cm$^2$. For the first sequences only small variations of the longitudinal resistivity were observed. However, with the application of subsequent pulse trains the magnitude of the effect increased. This training-like behaviour may be associated with the motion and pinning of AFM domain walls in the sample. After 1600 pulse trains a constant resistance change of $\Delta\rho_{long}/\overline{\rho} = 2.5\%$ induced by 100 pulse trains was reached, which is an order of magnitude higher than what is observed for CuMnAs[4].

To check the origin of these changes, the transversal resistivity of the sample was measured and also showed reproducible pulse current-induced changes (Fig. 4b). The increase of the transversal resistivity induced by 100 pulse trains amounted to $\rho_{xy} = 0.5$ μΩ cm.

Based on these numbers the identification of the longitudinal and transversal resistivities with the AMR and PHE can be verified: If both effects originate from the same anisotropic electron scattering, they have to be related by Eq. (2). We assume a switching of the Néel vector in parts of the

sample corresponding to a change of $\phi$ in Eq. (2) from +45° to −45°, i.e.,

$$\Delta\rho_{xy} = \rho_{xy}(+45^o) - \rho_{xy}(-45^o) = \Delta\rho_{long}, \quad (3)$$

Thus we find that $\rho_{xy} = 0.5$ μΩ cm corresponds again to $\Delta\rho_{long}/\overline{\rho} = 2.5\%$. This consistency of the longitudinal and transversal resistivities provides strong evidence for an intrinsic electronic origin of the pulse current-induced changes of the magnetoresistance signals.

After two more pulse current direction reversals reproducing the previous behaviour of the sample, the pulse current direction was kept along [100] for 800 additional pulse trains. This resulted in a sign reversal of the PHE, which according to Eq. (2) corresponds also to a sign change of the angle $\phi$ between the Néel vector and the current direction. Although a small offset of the transversal voltage due to e. g. imperfections of the patterned structure is possible, the magnitude of the transversal voltage measured with both signs can only be explained by a switching of the Néel vector. As no saturation of the read-out signals without destroying the samples could be reached, we can conclude that $\phi$ was switched from +45° to −45° for the most part of the sample, but not everywhere. After about 300 pulse trains along the [100] direction a beginning saturation of the PHE resistivity appeared, but was not completed when after 500 additional pulse trains the sample broke. A maximum transversal resistivity of $\rho_{xy} = 1.25$ μΩ cm was reached, which based on Eq. (3) corresponds to an AMR of 6.25%. This is one of the largest AMR ratios found in metallic magnetic thin films, and its size bodes well for easy read-out of the antiferromagnetic state as necessary for device applications. While small variations exist between samples, we observe consistently larger AMR effects for pulse currents along the [100] than for the [110] directions.

**Origin of the AMR**. To understand the origin of the magnetoresistance effects, we calculated the AMR of single domain Mn$_2$Au assuming a complete 90° switching of the Néel vector. In general, the AMR originates from effects of spin-orbit coupling on the band structure[22] and from scattering from an extrinsic disorder potential[23]. Incorporating the effects of realistic disorder in the calculations, two types of disorder were considered: off-stoichiometry and inter-site swapping between Mn and Au atoms. Experimentally, the former was analyzed by energy dispersive x-ray spectroscopy (EDX) of 500 nm thick Mn$_2$Au films resulting in a stoichiometry of 66.2 ± 0.3% Mn and 33.8 ± 0.3% Au, which indicates a slight Au excess. In addition, a small degree of inter-site disorder is to be expected, but its quantification is experimentally not accessible. Thus we simulated a slight excess of Au randomly distributed over the Mn sites and random Mn–Au swapping.

We calculated the AMR for two crystal directions of the Néel vector, AMR$_{100}$ and AMR$_{110}$ from ab initio theory. We employ the fully relativistic Dirac tight-binding linear-muffin tin orbital (FRD-TB-LMTO) density functional theory (DFT) combined with the Kubo-Greenwood formula for the longitudinal conductivity (see refs. [24–26]):

$$\sigma_{xx} \sim \sum_{\mathbf{k}} \mathrm{Tr}\left(\mathrm{Im}\overline{g}_{\mathbf{k}} V_{x,\mathbf{k}} \mathrm{Im}\overline{g}_{\mathbf{k}} V_{x,\mathbf{k}}\right) + \mathrm{v.c.}, \quad (4)$$

where $V_{x,\mathbf{k}}$ is the velocity operator labeled by the Cartesian coordinate $x$, and wavenumber $\mathbf{k}$, $\mathrm{Im}\overline{g}_{\mathbf{k}}$ is the auxiliary Green function within the TB-LMTO formalism[26], all evaluated at the Fermi level, $\overline{g}$ denotes the configuration averaging in the presence of disorder, and v.c. are vertex corrections[27]. We treat the scattering in the chemically disordered Mn$_2$Au within the coherent potential approximation (CPA). To reveal the role of

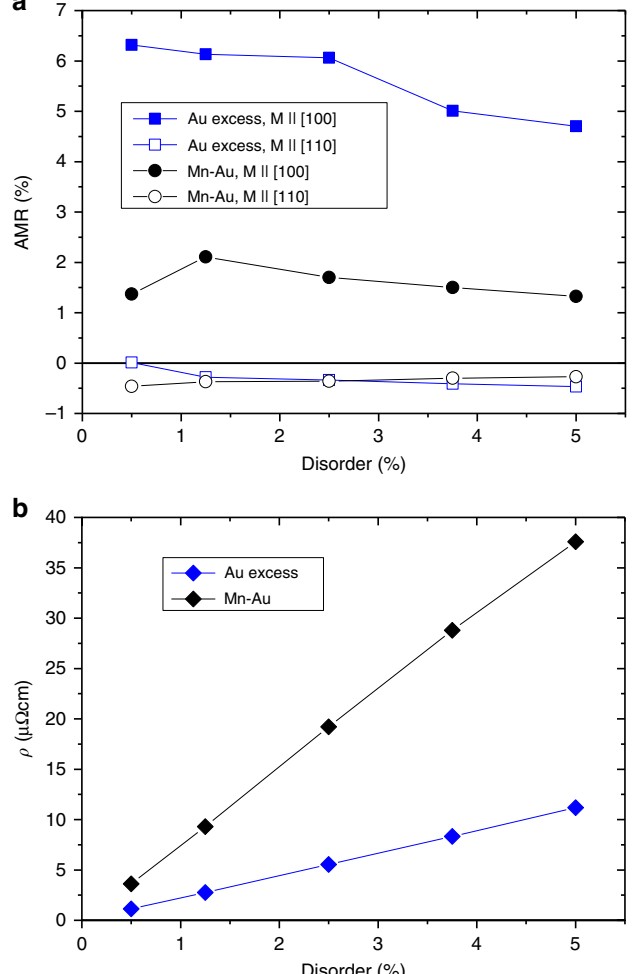

**Fig. 5** Calculated AMR. **a** Calculated AMR of $Mn_2Au$ for different degrees of disorder due to Au excess and due to Mn–Au site swapping with dependence on the Néel vector orientation. **b** Calculated residual resistivities of $Mn_2Au$ for different degrees disorder due to Au excess and due to Mn–Au site swapping

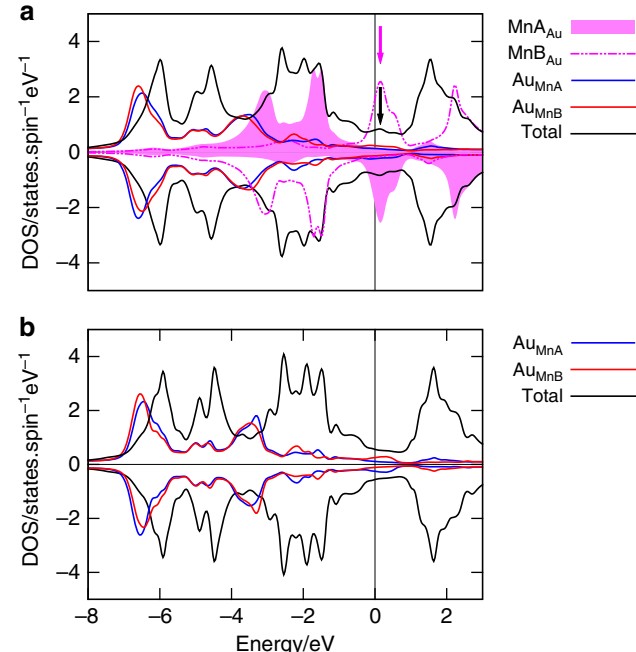

**Fig. 6** Atom and spin resolved DOS as calculated ab initio. **a** Total and disordered atom resolved DOS of $Mn_2Au$ with 5% Mn–Au swapping. The magenta (black) arrow marks the virtual bound state in the Mn on Au site (total) density of states. **b** DOS of $Mn_2Au$ with 5% excess of Au

the AMR contribution from spin-orbit coupling effects in the band structure, we also calculated the AMR of $Mn_2Au$ without chemical disorder in the constant relaxation time approximation (RTA) (see Methods section). The theoretical AMR is obtained by calculating the resistivity tensor $\rho = \sigma^{-1}$ from Eq. (4) and substituting it into Eq. (1).

Figure 5a shows the AMR results obtained within the CPA for different degrees of disorder in $Mn_2Au$. Large AMR values between 5 and 6%, which are consistent with our experiments, were obtained for Au excess generated scattering. Large AMR values consistent with our experiments were obtained for small degrees of disorder reaching a maximum value of 6.3% for 0.5% excess of Au. Moreover, we obtain $AMR_{100} > AMR_{110}$, which reproduces the experimentally observed trend for the two crystalline directions. The corresponding calculated residual resistivities as shown in Fig. 5b are consistent with the experimentally obtained values ($\approx 8\,\mu\Omega\,cm$)[17], corroborating the relevance of the simulated type of disorder.

The two types of disorder have different effects on the density of states (DOS) calculated within the FRD-TB-LMTO+CPA[24,26]. In the case of the Mn–Au swapping disorder, Mn located at the Au sites creates a virtual bound states at the Fermi level as shown in Fig. 6a, which increases the resistivity (Fig. 5b). However, such

states are not created by Au excess as shown in Fig. 6b. Thus these virtual bound states (VBS) are the dominating origin of the disorder induced resistivity increase but not of the large AMR. We note that the VBS are observed for the all simulated concentrations of Mn–Au swapping and for the sake of clarity we choose in Fig. 6 the largest simulated disorder strength since it leads to the better visibility of the VBS peak in the total density of states.

Instead of considering a specific type of disorder, scattering can also be treated within the RTA by broadening the imaginary part of the complex energy $\mathrm{Im}z$, which introduces unspecific lifetime effects on the electronic states, but otherwise keeps the band structure unperturbed by the disorder. With $\mathrm{Im}z \sim 13\,meV$ we achieved a residual resistivity $\simeq 10\,\mu\Omega\,cm$, which corresponds to the experimental low temperature values. Within this approximation we obtained $AMR_{100} \sim 1\%$ and $AMR_{110} \sim 0.1\%$. As these RTA values are significantly smaller than the CPA values, they indicate an extrinsic origin of the experimentally observed large AMR related to the Au excess on Mn sites as described by the CPA.

To further elucidate the effect of this Au excess on the electronic structure and related origin of the large scattering contribution to the AMR, we calculated the Bloch spectral function $A_{\mathbf{k}} = -\frac{1}{\pi}\mathrm{Im}\overline{G}_{\mathbf{k}}$. Here the physical Green function $\overline{G}$ is related to the auxiliary Green function $\overline{g}$ in Eq. (4)[26] and is calculated within the CPA along the high symmetry lines in the Brillouin zone of $Mn_2Au$ with 0.5% excess of Au. Since the vertex correction turned out to be negligible in $Mn_2Au$ with excess of Au, the Bloch spectral function represents an appropriate visualization of the disordered effects. As shown in Fig. 7 the spectral weight at the Fermi level along $M\Gamma$ line increases when rotating the Néel vector from || [100] (Fig. 7a) to || [010] (Fig. 7b). This suggests that the sharper spectral weight in the direction parallel as compared to perpendicular to the current direction effectively increases the corresponding relaxation time and thus reduces the resistivity. Please note that this picture remains

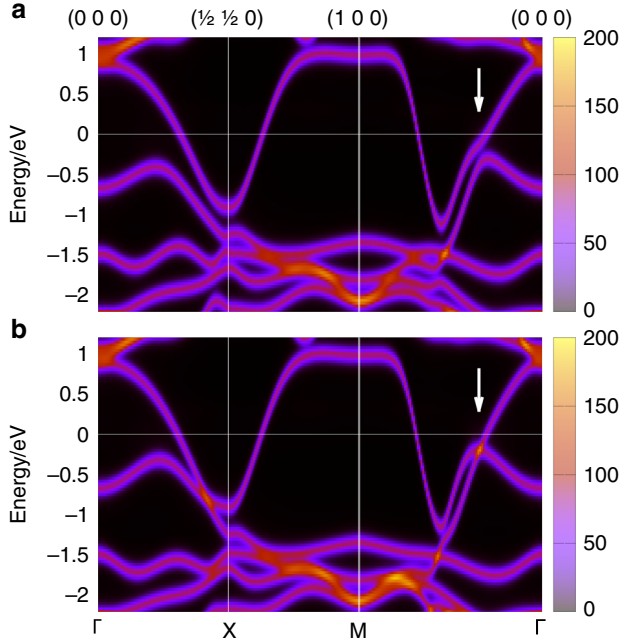

**Fig. 7** Bloch spectral function calculated within the CPA for 0.5%-Au-rich $Mn_2Au$ for two orientations of the Néel vector: **a** ∥ [100] and **b** ∥ [010]. The white arrows mark the major changes of the spectral function induced by the Néel vector rotation

qualitatively unchanged also for higher degrees of disorder, which we explicitly verified for 5% Au excess. This together with the k-space anisotropy of the MΓ band around the Fermi level can explain the high value of $AMR_{100}$ experimentally observed for Au-rich $Mn_2Au$.

## Discussion

In-plane switching of the Néel vector in the antiferromagnetic metal $Mn_2Au$ by current pulses was realized using intrinsic spin-orbit torques. Consistent measurements of the AMR and PHE showed pulse current direction dependent reversible changes, providing direct evidence for Néel vector switching. Easy read-out of the switching is provided by a large amplitude of the AMR of more than 6%, which is more than an order of magnitude higher than previously observed for other antiferromagnetic systems and one of the highest AMR amplitudes found for metallic magnetic thin films. We can reproduce the magnitude of the effect theoretically by including realistic disorder and, in particular, find the same dependence of the amplitude on the crystallographic directions in the experiment as in the calculation. With the basic principles of writing and read-out demonstrated, combined with a theoretical understanding of the underlying spin-orbit torques, and the large magnetoresistive effects, the metallic compound $Mn_2Au$ is a prime candidate to enable future AFM spintronics.

## Methods

**MAE calculation**. The MAE of chemically ordered $Mn_2Au$ was calculated using the FLAPW (full potential linearized augmented plane wave)+GGA (Generalized Gradient Approximation) method in combination with the magnetic force theorem[28]. We found the in-plane MAE: $E_{[100]} - E_{[110]} \lesssim |\pm 10|$ μeV per formula unit. However, this method is suitable only for fully ordered crystal structures. Thus to realistically describe our experiments we calculated the MAE within the TB-LMTO +CPA also for disordered $Mn_2Au$. For all concentrations of the simulated types of disorder we obtained the out-of-plane MAE $E_{[001]} - E_{[100]} \simeq 2.9$ meV. The in-plane MAE calculation gives $E_{[100]} - E_{[110]}$ from $-1$ to $-3$ μeV per formula unit at the resolution limit of our methods. While the out-of-plane MAE is similar to the reported value for chemically ordered $Mn_2Au$[29], the in-plane MAE of the disordered material has opposite sign. Independent from the sign the tiny value of the in-plane MAE at the resolution limit of the calculations is consistent with the

experimental observation that by current switching the [100] as well as the [110] Néel vector orientation could be stabilized in disordered $Mn_2Au$.

**AMR calculation**. To calculate the AMR of $Mn_2Au$ ab initio we employed the FRD-TB-LMTO+CPA method in combination with the Kubo formula[24–26]. s-type, p-type, and d-type orbitals were included in the basis and the LSDA (local spin density approximation) and the Vosko-Wilk-Nusair exchange-correlation potential parametrization[30] were used. The ground-state magnetization and density of states was reproduced consistently with a previous report[31]. In the transport calculations we used up to $10^{10}$ k points in the Brillouin zone and for the CPA residual resistivity calculations we set the imaginary part of the complex energy to 0.13 meV. In the RTA, the imaginary part of the complex energy Im$z$ in the Bloch spectral function $A_{n,\mathbf{k}}(z)$ was approximated by a finite isotropic k-independent relaxation time. In the CPA, the effective medium was constructed corresponding to a random averaging of the occupancies of the disordered sites. The CPA Bloch spectral function is anisotropic and k-dependent and determined by the effective medium potential.

We compared the RTA and the CPA derived calculated resistivities with the corresponding experimental values, from which we concluded that the level of disorder in our samples is at least 0.5%. Resistivity calculations for lower disorder values around, e.g., 0.1% correspond to 10–20 times smaller residual resistivities than observed in experiment and are thus not relevant in the framework of our manuscript. Nevertheless, we also calculated $AMR_{100}$ for 0.1% of Au excess resulting in a value of $(3.7 \pm 1.5)$%. The large error of this calculation is due to the increased number of iterations and k points needed for calculations with such a tiny degree of disorder.

**Data availability**. The relevant data are available within the article or from the authors on reasonable request.

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

## Acknowledgements

This work is supported by the German Research Foundation (DFG) through the Transregional Collaborative Research Center SFB/TRR173 Spin+X, Projects A03 and A05. J.S., L.S., O.G., and T.J. acknowledge the support of the Alexander von Humboldt Foundation, the ERC Synergy Grant SC2 (No. 610115), the Ministry of Education of the Czech Republic Grant No. LM2015087, and the Grant Agency of the Czech Republic grant no. 14-37427G, L.S. acknowledges support from the Grant Agency of the Charles University, no. 280815. Access to computing and storage facilities owned by parties and projects contributing to the National Grid Infrastructure MetaCentrum provided under the program Projects of Large Research, Development, and Innovations Infrastructures (CESNET LM2015042), is greatly appreciated. This work was supported by The Ministry of Education, Youth and Sports from the Large Infrastructures for Research, Experimental Development and Innovations project IT4 Innovations National Supercomputing Center LM2015070. The work of I.T. was supported by the Czech Science Foundation (Grant No. 14-37427G).

## Author contributions

S.Y.B. and M.J. mainly wrote the paper and performed the transport measurements; L.S. performed the AMR and anisotropy calculations and wrote the corresponding part of the manuscript, I.T. developed the codes for the transport calculations, S.Y.B. and A.A.S. prepared the samples, H.-J.E., M.K., O.G., T.J., I.T., and J.S. discussed the results and contributed to the writing of the manuscript; M.J. coordinated the project.

## Additional information

**Competing interests:** The authors declare no competing financial interests.

