## [Peer Review File · Nature Communications]

Reviewers' comments:

Reviewer #1 (Remarks to the Author):

The paper by Bodnar et al. reports on the current induced changes in resistivity of antiferromagnetic Mn(2)Au sputtered films (75 nm). The films were patterned into a star-structure (Fig. 1) which allows for applying currents along two perpendicular directions and measuring the longitudinal and transverse resistivities of the films. The authors follow the recipe of previous switching demonstrations in antiferromagnetic CuMnAs [ref. 4] and attribute the observed changes in Mn(2)Au resistivity to the current induced Néel vector switching probed by anisotropic magnetoresistance (AMR) effect. These observations could also report on one of the largest AMR (> 6%) found in metallic magnetic thin films and contribute significantly to the emerging field of antiferromagnetic spintronics. This new field is very promising for technology and has a lot to be demonstrated for the first time. However I have the following concerns about this manuscript:

-It is not clear why trains of current pulses were used in this experiment. What is a difference between 100 of 1 ms pulses with 10 ms delay, 100 of 0.5 ms pulses with 10 ms delay, 100 of 1 ms pulses with 20 ms delay and one 100 ms pulse? The effects of pulse and delay duration need to be clarified.

-There seems to be a correlation between the applied current density and the magnitude of the resistance change (Fig. 2) but other claims, like the training-like behavior or reaching a constant 2.5% change induced by 100 pulse (both in Fig. 3a), are not particularly justified. In general, more statistics is needed to describe the effects of current pulses. One can be much more quantitative about the current density effects on the resistance changes in Fig. 2.

-I would suggest reaching a real saturation in Fig. 2 because none of the reported single polarity measurements was long enough to do that.

-Figs. 3a and 3b should show CORRELATED transversal and longitudinal resistivities (measured simultaneously). Especially in view of using such correlations to support an intrinsic electronic origin of the pulse current induced changes of the magnetoresistance signals.

-The authors speculate a lot about the Néel vector orientation in their samples while no magnetic characterization was provided in support of their claims. For example: they assume a switching of the Néel vector that corresponds to a change of ϕ in equation (2) from +45 deg to -45 deg. This is done without seeing a clear saturation of measured longitudinal/transverse resistances. On another occasion the authors claim that after additional 800 pulse trains the Néel vector has

switched in the majority of the sample vs only parts of the sample after fewer pulse trains. What majority and parts of the sample were considered here and why the sign of transverse resistivity could be used to identify these part is not clear.

-It seems that star structures with two crystallographic orientations of Mn(2)Au films were used in the experiments: one which allows for applying currents along [110] and [1-10] directions, and one - along [100] and [010]. Is that correct? The authors should be more clear about samples being measured. Indication of crystallographic orientation(s) in Fig. 1 would be helpful.

Reviewer #2 (Remarks to the Author):

The description of the preparation of the sample and calculation processes are too unclear to understand for most readers. I will not recommend this article to publish in Nature Communications. There are some basic problems need to be clarified for this article before any serious considerations.

1. there are no X-ray photoemission spectroscopy compositional analysis, TEM image, and temperature dependent M (H) loops for your experimental sample.
2. there is no structural parameter for theoretical calculation such as: supercell size, magnetization direction for your MAE calculation. According to personal experience, the defect and heterostructure are two complicated problems in first-principles calculation. How do the authors simplify and discuss in the theoretical calculation compare with the experimental data should explicitly state. In the method section, authors only calculate the MAE value of bulk Mn₂Au and the same value with previous reports in reference²³. This kind of the discussion does not provide any useful and sensible information from fundamental physics. Authors need to discuss about the effect of disorder on MAE instead.
- 3, How do authors construct the small disorder structure of their AMR calculation? The reference 25 focus on the disordered Co-based Heusler alloys but not Mn₂Au case.
4. According PRL 113, 157201 (2014), the switch current should be around 10^8 – 10^9 A/cm². In this paper, the current density is only 10^7 A/cm². Why can they re-orientate magnetic moment of Mn₂Au with such a small current density?

In summary I do not think current version is suitable to publish.

Reviewer #3 (Remarks to the Author):

The authors have demonstrated Mn₂Au as a new material for reliable and reproducible switching using current pulses and read-out by magnetoresistance measurements. Evidences from both experiments and theoretical calculations (e.g., anisotropic magnetoresistance) have been provided. This finding is interesting and could be useful in spintronics devices.

However, the theoretical results raise a few questions:

1. The calculation of magnetic anisotropy energy is not clear. A few details are required, such as the k-points sampling, energy cut-off, energy convergence, the number of atoms in the supercell and spin-orbital coupling. These settings are important since the authors reported MAE is about 1 μeV , at the resolution limit of classical ab initio calculations.

This value is different from the mentioned literature (Ref. 23), where the in-plane MAE constant is about 10 μeV per formula unit and the uniaxial anisotropy constant is as large as 2440 μeV per formula unit. It is not clear what causes this difference.

2. The experimental samples (500 nm thick films) have a slight Au excess. Will Au excess significantly affect the magnetic anisotropy?

3. How large is AMR for the sample without any defects (i.e, Au excess, and Mn-Au site swapping)? Figure 4(a) shows AMR ([100]) increases with a smaller Au excess and about 6.5% is expected for 0% disorder. But in the case of Mn-Au swapping, this value is expected to be less than 1.5%.

It is also not clear why the curve ([100]) of Mn-Au swapping has a peak in about 1.25% disorder. Is it because the random sampling is not large enough in statistics?

4. The calculated maximal AMR (6.3%) is in good accordance with the measured one (6.25%). This indicates a slight and random Au excess is the major reason for the large AMR. It would be better to add the analysis from electronic structures and spin-orbital coupling effect. This would provide insight for the origin of the large AMR.

5. Minor points

(a). Explain the definition of the denominator in Formula (1)

(b). On page 7, "The ground-state magnetization and density of states was reproduced consistently with a previous reports [3, 12]." Was this ground-state calculation performed on the

bulk Mn₂Au without defects? The results can be provided as supplemental materials to support the accuracy of theory calculations.

I suggest a further consideration of this manuscript after the issues listed above have been well addressed.

Responses to Reviewer Comments:

We would like to thank the referees who's criticism helped to improve our manuscript considerably. All issues raised are discussed in detail in the point-by-point response below. Additionally, in the revised version of our manuscript the new sections are printed in blue. Also below citations of new sections from the manuscript are printed in blue.

Reviewer 1:

- *The paper by Bodnar et al. reports on the current induced changes in resistivity of antiferromagnetic Mn(2)Au sputtered films (75 nm). The films were patterned into a star-structure (Fig. 1) which allows for applying currents along two perpendicular directions and measuring the longitudinal and transverse resistivities of the films. The authors follow the recipe of previous switching demonstrations in antiferromagnetic CuMnAs [ref. 4] and attribute the observed changes in Mn(2)Au resistivity to the current induced Néel vector switching probed by anisotropic magnetoresistance (AMR) effect. These observations could also report on one of the largest AMR (> 6%) found in metallic magnetic thin films and contribute significantly to the emerging field of antiferromagnetic spintronics. This new field is very promising for technology and has a lot to be demonstrated for the first time. However I have the following concerns about this manuscript:*

It is not clear why trains of current pulses were used in this experiment. What is a difference between 100 of 1 ms pulses with 10 ms delay, 100 of 0.5 ms pulses with 10 ms delay, 100 of 1 ms pulses with 20 ms delay and one 100 ms pulse?

To clarify the reason for the application of pulse trains instead of the previous short discussion of heating effects in the Methods section, we now discussed this issue extensively in the main text of the revised manuscript using the following text:

Switching and Read-Out. For technical reasons the applicable pulse length for switching was limited to a minimum of 1 ms. This allows us using an oscilloscope to monitor the time dependent sample resistivity during the application of the current pulses (Fig. 2). By comparison with the temperature dependence of the resistivity of our Mn₂Au thin films obtained from separate measurements, this allows to study the current pulse induced heating effects. As shown in Fig. 2 significant

2

heating with local temperatures up to 300 °C is obtained for the highest possible current densities not resulting in sample destruction.

Current induced Néel vector switching was only observed applying pulse current densities associated with notable heating. As single pulses resulted in very small changes of the read-out signals only, trains of 100 current pulses with a pulse length of 1 ms and a delay between the pulses of 10 ms were applied. This relatively long delay between the pulses was chosen to avoid the destruction of the sample by excessive accumulated heating.

- *The effects of pulse and delay duration need to be clarified.*

The measurement delay of 10 s after the application of a pulse train ensured that the sample was in an equilibrated condition when probed. We made this issue clearer by adding the following statement to the text of the manuscript:

As after a pulse train thermal relaxation behaviour on a time scale of 1 s after was observed, the read-out was performed with a delay of 10 s. The magnetic state of the sample probed this was long-term stable, i. e. no changes of the read-out signals were observed within the probed time scales of up to one hour.

Systematic studies of the effects of pulse and delay duration within the pulse trains are very challenging due to the above mentioned requirement to switch the sample close to its destruction limit and go beyond the scope of this proof-of-concept work for spin torque switching in Mn₂Au.

- *There seems to be a correlation between the applied current density and the magnitude of the resistance change (Fig. 2) but other claims, like the training-like behavior or reaching a constant 2.5% change induced by 100 pulse (both in Fig. 3a), are not particularly justified. In general, more statistics is needed to describe the effects of current pulses. One can be much more quantitative about the current density effects on the resistance changes in Fig. 2.*

The previous Fig. 2 is now Fig. 3. As the data shown in Fig. 4 demonstrates using a constant pulse current density typically results in an increase of the corresponding read-out signals during the first several hundred pulse trains. Thus a systematic study of the correlation between applied current density and magnitude of the resistance change does require a large number of current pulse train reversals for each current density to allow for a separation of the training and current pulse density related effect. Due to the above explained fragility of our samples we are at the moment unable to supply such a certainly desirable investigation in a systematic way going beyond Fig. 3. While certainly of interest for a future study beyond the scope of the current work, we believe that this does not affect the general statements of our paper about the possibility to switch the Néel vector and about the associated large AMR.

- *I would suggest reaching a real saturation in Fig. 2 because none of the reported single polarity measurements was long enough to do that.*

The previous Fig. 2 is now Fig. 3. We have studied further samples and we show now in the new inset of Fig. 4a an example that demonstrates that in principle it is possible to apply current pulses until a large degree of saturation of the induced longitudinal resistivity changes is obtained. We now discuss this issue in the manuscript by adding the following statement: The inset of Fig. 4a shows that it is possible to apply current pulses until a large degree of saturation of the induced longitudinal resistivity changes is obtained. However, this is achieved for current densities that result for prolonged injection of pulses in the destruction of the samples. Thus to allow for an increased number of current pulse cycles with different polarities this regime has to be avoided. The current density at which eventually the sample is destroyed varies by a factor of ≈ 2 from sample to sample.

- *Figs. 3a and 3b should show CORRELATED transversal and longitudinal resistivities (measured simultaneously). Especially in view of using such correlations to support an intrinsic electronic origin of the pulse current induced changes of the magnetoresistance signals.*

The previous Fig. 3 is now Fig. 4 (with a new inset added). During the application of the pulse trains all other instruments have to be disconnected from the sample due to the associated large voltages. We solved this problem using a scanner unit which connected and disconnected the current source and nanovoltmeter used for probing. However, this scanner unit does not provide enough independent channels for alternating transversal and longitudinal resistivity measurements after each pulse train. However, the measurements shown in Fig. 4b was performed directly after the measurement shown in Fig. 4a (after manual reconfiguration of the connection of the measurement devices). As we know that the samples are stable with no changes over hours, the shown measurements are indeed correlated.

- *The authors speculate a lot about the Néel vector orientation in their samples while no magnetic characterization was provided in support of their claims. For example: they assume a switching of the Néel vector that corresponds to a change of ϕ in equation (2) from +45 deg to -45 deg. This is done without seeing a clear saturation of measured longitudinal/transverse resistances. On another occasion the authors claim that after additional 800 pulse trains the Néel vector has switched in the majority of the sample vs only parts of the sample after fewer pulse trains. What majority and parts of the sample were considered here and why the sign of transverse resistivity could be used to identify these part is not clear.*

In principle direct imaging of the Néel vector orientation could be possible using XMLD-PEEM. However, our spectroscopic XMLD investigations of the Mn₂Au thin films showed only a very small XMLD effect generated by a spin-flop transition in a 70T magnetic field (A. Sapoznik et al., Phys. Status Solidi 11, 1600438 (2017), attached). This and a possibly very small AFM domain size will make XMLD-PEEM investigations very challenging. Nevertheless, we will attempt this in future. However, such detailed insight is not required for the general statements of our paper about the possibility to switch the Néel vector and about the associated large AMR.

Concerning the relation between the observed sign reversal of the PHE signal and our conclusion that this is an indication of Néel vector switching in the majority of the sample, it seems that our explanations given in the manuscript were not sufficiently clear. Thus we modified the corresponding text in the following way:

This resulted in a sign reversal of the PHE, which according to eq. (2) also corresponds to a sign change of the angle ϕ between the Néel vector and the current direction. Although a small offset of the transversal voltage due to e. g. imperfections of the patterned structure is possible, the magnitude of the transversal voltage measured with both signs can only be explained by a switching of the Néel vector. As no full saturation of the read-out signals without destroying the samples could be reached, we can conclude that ϕ was switched from +45° to -45° for the most part of the sample, but not everywhere.

- *It seems that star structures with two crystallographic orientations of Mn(2)Au films were used in the experiments: one which allows for applying currents along [110] and [1-10] directions, and one - along [100] and [010]. Is that correct? The authors should be more clear about samples being measured. Indication of crystallographic orientation(s) in Fig. 1 would be helpful.*

Apparently our description of the samples was not clear enough. So we clarified this issue by adding additional panels to Fig. 1, which show the crystallographic directions involved. Additional to the corresponding figure caption the following text was added to the manuscript:

Our Al₂O₃ (Substrate)/Ta(10nm)/Mn₂Au(75 nm)/Ta(3 nm) samples were prepared by radio frequency sputtering from a single stoichiometric target and structurally and magnetically characterized as described elsewhere [17]. By x-ray diffraction we demonstrated that the Mn₂Au thin films grow with the (001) axis perpendicular to the thin film surface. The in-plane orientation is given by the epitaxial relation with the Ta(001) buffer layer, which results in the [100]-direction of the Mn₂Au thin films aligned parallel to the [100]-direction of the epitaxial Ta layer. These samples were patterned into a star-structure as shown in Fig. 1. ... Depending on the in-plane orientation of the patterned structure, the pulse currents could be sent along different crystallographic directions, i. e. along [100] or along [110] (45° in-plane rotation of the star pattern).

Reviewer 2:

- *The description of the preparation of the sample and calculation processes are too unclear to understand for most readers. I will not recommend this article to publish in Nature Communications. There are some basic problems need to be clarified for this article before any serious considerations.*

We thank the referee for this comment and have extended our explanations. Concerning preparation we added the following statement:

Our Al_2O_3 (Substrate)/Ta(10 nm)/ Mn_2Au (75 nm)/Ta(3 nm) samples were prepared by radio frequency sputtering from a single stoichiometric target and structurally and magnetically characterized as described elsewhere [17]. By x-ray diffraction we demonstrated that the Mn_2Au thin films grow with the (001) axis perpendicular to the thin film surface. The in-plane orientation is given by the epitaxial relation with the Ta(001) buffer layer, which results in the [100]-direction of the Mn_2Au thin films aligned parallel to the [100]-direction of the epitaxial Ta layer.

- *1. there are no X-ray photoemission spectroscopy compositional analysis, TEM image, and temperature dependent $M(H)$ loops for your experimental sample.*

We performed a compositional analysis of several of our samples by energy dispersive x-ray spectroscopy (EDX) as described in our manuscript:

“Experimentally, the former (Off-stoichiometry) was analyzed by energy dispersive x-ray spectroscopy (EDX) of 500 nm thick Mn_2Au films resulting in a stoichiometry of 66.2 ± 0.3 % Mn and 33.8 ± 0.3 % of Au, which indicates a slight Au excess.” EDX is a very reliable method if the investigated thin film sample is sufficiently thick (like 500nm of Mn_2Au) so that it corresponds well to the involved quantification algorithms which assume a homogeneous semi-infinite sample. Thus we do not see any need for an XPS analysis that would yield the same information.

Concerning structural analysis we used 4-circle x-ray diffraction and electron diffraction instead of TEM to establish the epitaxial growth relations and thin films order as mentioned above. We hope that the new statements as mentioned above now make clear that these investigations exist.

Concerning $M(H)$ loops: Those were measured in a SQUID magnetometer in order to estimate if a significant amount of uncompensated moments is present in our antiferromagnetic samples, which is not the case as shown in the figure on the right.

- *2. there is no structural parameter for theoretical calculation such as: supercell size, magnetization direction for your MAE calculation. According to personal experience, the defect and heterostructure are two complicated problems in first-principles calculation. How do the authors simplify and discuss in the theoretical calculation compare with the experimental data should explicitly state. In the method section, authors only calculate the MAE value of bulk Mn_2Au and the same value with previous reports in reference23. This kind of the discussion does not provide any useful and sensible information from fundamental physics. Authors need to discuss about the effect of disorder on MAE instead.*

We simulate the disorder effects by employing an *ab initio* coherent potential approximation rather than a supercell method, as now explained in more detail in the section **Origin of the AMR** as well as in the **Methods** part of the manuscript (new statements printed in blue in the revisions version of our manuscript). Since the samples are quite thick (100 nm) we calculate the MAE and AMR for bulk disordered Mn_2Au . We obtained a small in-plane MAE of the order of microelectronvolts, which is at the resolution limit of the *ab initio* calculations and a larger out-of-plane MAE, which is consistent with the experimental evidence for an easy (001) plane. In the **Methods** section of the revised manuscript we added more details

on the MAE calculations and considered the MAE for the disordered cases. As a background information we summarize in the following table results obtained by different approaches.

Ab initio method	MAE method	Disorder	E[100]-E[001] (meV per f.u.)	E[100]-E[110] (meV per f.u.)	k points
FLAPW LDA Wien Ref	MFT*	Clean	-2,42	0,02	6875
FLAPW GGA Fleur	MFT	Clean	-2,12	-0,001	24576
LMTO + LSDA	TE	Clean	-2,89	-0,008	13824
LMTO+LSDA+CPA	TE	Au x=2.5	-2,88	-0,003	5832
LMTO+LSDA+CPA	TE	Mn-Au x=2.5	-2,88	-0,003	5832

Method legend: MFT = magnetic force theorem, TE = total energies

* According to Ref. Schick et al, Phys. Rev. B (2010)

- 3, How do authors construct the small disorder structure of their AMR calculation? The reference 25 focus on the disordered Co-based Heusler alloys but not Mn₂Au case.

We employed the Coherent Potential Approximation (CPA) constructing an effective medium. We include in the **Origin of the AMR** as well as in the **Methods** section more details concerning our *ab initio* calculation of the electronic structure, conductivity, and AMR of disordered Mn₂Au (new statements printed in blue in the revisions version of our manuscript).

- 4. According PRL 113, 157201 (2014), the switch current should be around 10^8 – 10^9 A/cm². In this paper, the current density is only 10^7 A/cm². Why can they re-orientate magnetic moment of Mn₂Au with such a small current density?

The compared to the theoretical prediction relatively low necessary current density for switching is presumably related to the already above mentioned heating effects caused by the current pulses, which are not taken into account in the theory (see new Fig. 2). In the new version of our manuscript, additional to the new heating related part mentioned above, this is now considered by the following statement:

... 1.7×10^7 A/cm² and 1.8×10^7 A/cm² resulted in larger changes of the corresponding Hall voltages. Those current densities are about 1 order of magnitude smaller than predicted by Zelezny et al. [3,21], which could be related to thermal activation processes enabled by the above mentioned current pulse induced heating.

Furthermore, the reference Zelezny PRB 2016 reports current induced fields which depend on the calculation method used (0.22 (tight binding) or 1.98 (ab initio) mT/10⁷ A/cm²). These authors also use an effective spectral broadening of 0.0013 eV, which 10 times smaller than the value estimated for our samples. Also their estimation uses the in-plane MAE values reported by reference Schick, PRB, 2011, which might be different in the real samples, the values of the energy barriers, according to our calculations for the disordered Mn₂Au are almost one order of magnitude smaller.

Reviewer 3:

- *The authors have demonstrated Mn₂Au as a new material for reliable and reproducible switching using current pulses and read-out by magnetoresistance measurements. Evidences from both experiments and theoretical calculations (e.g., anisotropic magnetoresistance) have been provided. This finding is interesting and could be useful in spintronics devices.*

However, the theoretical results raise a few questions:

*The calculation of magnetic anisotropy energy is not clear. A few details are required, such as the k-points sampling, energy cut-off, energy convergence, the number of atoms in the supercell and spin-orbital coupling. These settings are important since the authors reported MAE is about 1 μeV , at the resolution limit of classical *ab initio* calculations.*

This value is different from the mentioned literature (Ref. 23), where the in-plane MAE constant is about 10 μeV per formula unit and the uniaxial anisotropy constant is as large as 2440 μeV per formula unit. It is not clear what causes this difference.

We now discuss in the **Methods** section of the revised version of our manuscript this issue in greater detail (new statements printed in blue in the revisions version of our manuscript). The very small value of the in-plane anisotropy depends on the very subtle differences in the electronic structure calculations and can be attributed to the different approximations used (we use Dirac equation in combination with TB-LMTO and CPA, while Ref. 23 uses the Schrödinger equation combined with FP LAPW). Furthermore, since the values are at the resolution limit of the *ab initio* calculations, revealing the detailed origin of the subtle differences is beyond the scope of the present manuscript. For additional background information please see the MAE table shown above.

- *2. The experimental samples (500 nm thick films) have a slight Au excess. Will Au excess significantly affect the magnetic anisotropy?*

We now discuss in the **Methods** section of the revised version of our manuscript this issue in greater detail (new statements printed in blue in the revisions version of our manuscript). Please see also our reply above (page 4, last item, and 5, table).

- *3. How large is AMR for the sample without any defects (i.e., Au excess, and Mn-Au site swapping)? Figure 4(a) shows AMR ([100]) increases with a smaller Au excess and about 6.5% is expected for 0% disorder. But in the case of Mn-Au swapping, this value is expected to be less than 1.5%.*

The results of the CPA calculations of disordered Mn₂Au cannot be directly interpolated to zero disorder, because zero disorder without scattering would yield an unphysical infinite conductivity. The best available method to determine the AMR of Mn₂Au without chemical disorder is the constant relaxation time approximation (RTA). In the new version of the manuscript we include the results of AMR calculations for Mn₂Au without chemical disorder within the relaxation time approximation. Please see the corresponding new statements printed in blue in the section Origin of the AMR in the revisions version of our manuscript.

- *It is also not clear why the curve ([100]) of Mn-Au swapping has a peak in about 1.25% disorder. Is it because the random sampling is not large enough in statistics?*

The AMR value depends on the subtle details of the electronic structure and is not related to the sampling statistics, since we use the CPA, not a supercell method as now explained in more detail in the manuscript. Please see the new statements printed in blue in the sections Origin of the AMR and Methods in the revisions version of our manuscript.

- *4. The calculated maximal AMR (6.3%) is in good accordance with the measured one (6.25%). This indicates a slight and random Au excess is the major reason for the large AMR. It would be better to add the analysis from electronic structures and spin-orbital coupling effect. This would provide insight for the origin of the large AMR.*

7

We included new calculations and a discussion of the electronic structure related origin of the large AMR in the revised version of our manuscript. Based on a new calculations of (i) the band structure related AMR, (ii) the density of states around Fermi level (new Fig. 6), and (iii) the anisotropy of the spectral function within CPA (new Fig.7) deeper insight into the origin of the large AMR could be provided as described in the **new statements printed in blue in the sections Origin of the AMR in the revisions version of our manuscript**. The new calculations confirm our original statement concerning the dominating extrinsic origin of the AMR in the Mn_2Au .

- 5. Minor points
(a). Explain the definition of the denominator in Formula (1)

In eq. 1 we replaced the denominator by

$$(\rho_{long}(\phi = 0^\circ) + \rho_{long}(\phi = 90^\circ))/2$$

- (b). On page 7, “The ground-state magnetization and density of states was reproduced consistently with a previous reports [3, 12].” Was this ground-state calculation performed on the bulk Mn_2Au without defects? The results can be provided as supplemental materials to support the accuracy of theory calculations.

The comparison of the electronic structure with the existing literature was done for the chemically ordered Mn_2Au , because to the best of our knowledge no study including disorder was published up to now. We also performed ground-state calculation for bulk Mn_2Au without defects (corresponding to the new Fig. 6). As a background information below we show the density of states of defect free Mn_2Au , which is consistent with the previous reports: Khmelevski, Appl. Phys. Lett. (2008); Wu, Adv. Funct. Mat. (2012); and Schick, Phys. Rev. B (2010).

Reviewers' comments:

Reviewer #1 (Remarks to the Author):

I believe the authors have responded adequately to my questions and the paper can now be published.

Reviewer #3 (Remarks to the Author):

The authors have added more details and results for this manuscript and it now looks clearer. However, there are some points needing further clarifications.

1. There are several confusing information regarding the MAE calculation. In the previous manuscript, the MAE is calculated by “FLAPW (Full Potential Linearized Augmented Plane Wave) method in combination with the GGA (Generalized Gradient Approximation)” and the value is 1 μeV per formula unit. The current manuscript used the same method and the value is not greater than 10 μeV per formula unit. But in the response letter, the authors claimed “we use Dirac equation in combination with TB-LMTO and CPA, while Ref. 23 uses the Schrödinger equation combined with FP LAPW”. These are confusing. I understand the authors might have used two methods to calculate the MAE for the chemically ordered system and the in-plane MAE is very small. Please check these statements.
2. The AMR ([100]) is about 1% in the RTA calculation of the chemically ordered system. This value is close to the Mn-Au curve and different from the Au excess curve (Figure 5a). The CPA method cannot calculate the zero disorder. But in the reality (please correct if not right), the AMR of Au excess should converge to the chemically ordered system when the concentration of Au excess decreases. Can the authors provide the data for the lower disorder (e.g., 0.1%). This is important to confirm that the CPA method describes the Au excess correctly.
3. Since AMR ([100]) is about 1% for the chemically ordered system, it is not surprising that the curve ([100]) of Mn-Au swapping has a peak. One would guess the curve of Au excess has a similar trend and a peak. Once again, the results for the lower disorder are important.
4. The experiments found “a maximum transversal resistivity of $\rho_{xy} = 1.25$ for an AMR of 6.25%.” Seen from Figure 5, this corresponds to a small disorder (“Large AMR values consistent with our experiments were obtained for small degrees of disorder reaching a maximum value of 6.3 % for 0.5 % excess of Au”). Why did the authors use 5% Mn-Au swapping and 5% excess of Au for the calculations of Figure 6 and 7? The picture for the high disorder might not be the

same as the low disorder, since there is a peak in the Mn-Au curve (Figure 5a).

Responses to Reviewer Comments:

We would like to thank the referee who's criticism helped to improve our manuscript considerably. All issues raised are discussed in detail in the point-by-point response below. Additionally, in the revised version of our manuscript the new sections are **printed in blue**. Also below citations of new sections from the manuscript are **printed in blue**.

Reviewer 3:

- *The authors have added more details and results for this manuscript and it now looks clearer. However, there are some points needing further clarifications.*

1. There are several confusing information regarding the MAE calculation. In the previous manuscript, the MAE is calculated by "FLAPW (Full Potential Linearized Augmented Plane Wave) method in combination with the GGA (Generalized Gradient Approximation)" and the value is 1 μeV per formula unit. The current manuscript used the same method and the value is not greater than 10 μeV per formula unit. But in the response letter, the authors claimed "we use Dirac equation in combination with TB-LMTO and CPA, while Ref. 23 uses the Schrödinger equation combined with FP LAPW". These are confusing. I understand the authors might have used two methods to calculate the MAE for the chemically ordered system and the in-plane MAE is very small. Please check these statements.

Indeed we used both FLAPW+GGA and TB-LMTO+CPA to calculate the MAE. However, the former approach can only be used for fully ordered crystal structures, whereas the latter allows to include disorder. In the revised version of our manuscript we clarified this issue by adding in the Methods section: "**However, this method is suitable only for fully ordered crystal structures. Thus to realistically describe ...**" As already stated in the manuscript, both methods, independent for the degree of disorder, resulted in tiny MAE values, which are at the resolution limit of our methods, which we determined to be 10 μeV in the FLAPW case. This resolution limit was determined based on variations of the number of the numerical iterations.

- *2. The AMR ([100]) is about 1% in the RTA calculation of the chemically ordered system. This value is close to the Mn-Au curve and different from the Au excess curve (Figure 5a). The CPA method cannot calculate the zero*

2

disorder. But in the reality (please correct if not right), the AMR of Au excess should converge to the chemically ordered system when the concentration of Au excess decreases. Can the authors provide the data for the lower disorder (e.g., 0.1%). This is important to confirm that the CPA method describes the Au excess correctly.

Please note that the CPA for a low concentration and the RTA for the chosen spectral broadening does not have to give the same value. In our calculation, the spectral broadening in RTA was chosen to reproduce the experimental resistivities. Both approximations converge (if the limit exists) independently (in the limit of zero chemical disorder in the case of the CPA, and in the limit of zero spectral broadening in the case of RTA) to the intrinsic zero disorder value. Here we compared the RTA and the CPA derived calculated resistivities with the corresponding experimental values, from which we concluded that the level of disorder in our samples is at least 0.5%. Resistivity calculations for disorder values around 0.1% correspond to 10-20 times smaller residual resistivities than observed in experiment and are thus not relevant in the framework of our manuscript. Nevertheless, we also calculated AMR[100] for 0.1% of Au-excess resulting in a value of (3.7+-1.5)%. The large error of this calculation is due to the increased number of iterations and k-points needed for calculations with such a tiny degree of disorder.

To clarify our motivation to use the CPA and the RPA, we added the following statements to the manuscript:

“Instead of considering a specific type of disorder, scattering can also be treated within the RTA by broadening the imaginary part of the complex energy $\text{Im } z$, which introduces unspecific lifetime effects on the electronic states, but otherwise keeps the band structure unperturbed by the disorder.”

“As these RTA values are significantly smaller than the CPA values, they indicate an extrinsic origin of the experimentally observed large AMR related to the Au excess on Mn-sites as described by the CPA.”

- *3. Since AMR ([100]) is about 1% for the chemically ordered system, it is not surprising that the curve ([100]) of Mn-Au swapping has a peak. One would guess the curve of Au excess has a similar trend and a peak. Once again, the results for the lower disorder are important.*

As explained above, we now calculated AMR[100] for 0.1% of Au-excess resulting in a value of (3.7+-1.5)%, which indicates a convergence of the CPA and RTA values to the same zero disorder limit somewhere around $\text{AMR} \cong 1\%$. However, please note that the corresponding low degrees of disorder are not consistent with the experimentally observed resistivities of our samples (which indicate realistic degrees of disorder above 0.5%).

- *4. The experiments found “a maximum transversal resistivity of $\rho_{xy} = 1.25$ for an AMR of 6.25%.” Seen from Figure 5, this corresponds to a small disorder (“Large AMR values consistent with our experiments were obtained for small degrees of disorder reaching a maximum value of 6.3 % for 0.5 % excess of Au”). Why did the authors use 5% Mn-Au swapping and 5% excess of Au for the calculations of Figure 6 and 7? The picture for the high disorder might not be the same as the low disorder, since there is a peak in the Mn-Au curve (Figure 5a).*

The dependence of the calculated AMR on the degree of order is not very strong for disorder levels in the considered range between 0.5% and 5%. We adapted the above mentioned statement to: *Large AMR values between 5 and 6%, which are consistent with our experiments, were obtained for Au-excess generated scattering.*

We also added the following statement:

Thus these virtual bound states (VBS) are the dominating origin of the disorder induced resistivity increase but not of the large AMR. We note that the VBS are observed for the all simulated concentrations of Mn-Au swapping and for the sake of clarity we choose in Fig. 6 the largest simulated disorder strength since it leads to the better visibility of the VBS peak in the total density of states.

As a background information (not included in the manuscript) please find below the side by side comparison of the DOS for the 0.5 and 5% disorder:

3

Figure 1 Left panel: 0.5% of disorder. Right panel: 5% of disorder.

We now show in the revised version of Fig. 7 the spectral function for 0.5% Au-excess. As a background information (not included in the manuscript) please find below again a comparison of the spectral functions obtained for 0.5% (new Fig. 7) and 5% (old Fig. 7) Au-excess.:

Figure 2 Left panel: 0.5% of Au excess. Right panel: 5% of Au excess. Note the larger smearing of the bands for higher disorder and the shift of the hotspot away from the Fermi level in the case of higher disorder.

The observed decrease of the AMR for higher Au excess can be attributed to the shift of the chemical potential away from the hot-spot of the spectral redistribution points. Thus we added the following statement to the manuscript:

Please note that this picture remains qualitatively unchanged also for the higher disorders which we explicitly verified for 5% Au-excess

Finally, we moved the statement about the vertex corrections from the Methods section into the main text: $\text{Im } g_{\mathbf{k}}$ is the auxiliary Green function within the TB-LMTO formalism [26], all evaluated at the Fermi level, g denotes the configuration averaging in the presence of disorder, and v.c. are vertex corrections [27].

4

By this we facilitate the detailed explanation of the use of the Bloch spectral function as a disorder visualization tool. The new sentences reads:

Here the physical Green function G is related to the auxiliary Green function g in Eq.4 [26] and is calculated within the CPA along the high symmetry lines in the Brillouin zone of Mn_2Au with 0.5% ~~5%~~ excess of Au. Since the vertex correction turned-out to be negligible in Mn_2Au with excess of Au, the Bloch spectral function represents an appropriate visualization of the disordered effects.

Reviewers' Comments:

Reviewer #3 (Remarks to the Author):

The authors have solved my questions.

One optional point:

Add the discussions on 0.1% of Au-excess, to indicate the convergence of the calculated methods.

Responses to Reviewer Comments:

Reviewer 3:

- *The authors have solved my questions. One optional point: Add the discussions on 0.1% of Au-excess, to indicate the convergence of the calculated methods.*

We followed the advice of the reviewer by adding the following statement from our previous response letter to the Methods (AMR calculation) section of our manuscript:

“We compared the RTA and the CPA derived calculated resistivities with the corresponding experimental values, from which we concluded that the level of disorder in our samples is at least 0.5 %. Resistivity calculations for lower disorder values around e.g. 0.1 % correspond to 10 – 20 times smaller residual resistivities than observed in experiment and are thus not relevant in the framework of our manuscript. Nevertheless, we also calculated AMR100 for 0.1 % of Au-excess resulting in a value of $(3.7 \pm 1.5) \%$. The large error of this calculation is due to the increased number of iterations and k-points needed for calculations with such a tiny degree of disorder.”